# Predictors of Early Continence Recovery Following Radical Prostatectomy, Including Transperineal Ultrasound to Evaluate the Membranous Urethra Length (CHECK-MUL Study)

**DOI:** 10.3390/diagnostics14080853

**Published:** 2024-04-21

**Authors:** Bara Barakat, Mustapha Addali, Boris Hadaschik, Christian Rehme, Sameh Hijazi, Samy Zaqout

**Affiliations:** 1Urology Centre, Albertusstraße 17, 41061 Möchengladbach, Germany; doc-sam@hotmail.de; 2Department of Urology and Pediatric Urology, Hospital Viersen, 41747 Viersen, Germany; 3Department of Urology, Hospital Siegen, 57076 Siegen, Germany; m.addali@klinikum-siegen.de; 4Department of Urology, University Hospital Essen, 45147 Essen, Germany; boris.hadaschik@uk-essen.de (B.H.); christian.rehme@uk-essen.de (C.R.); 5Department of Urology, Hospital Ibbenbüren, 49477 Ibbenbüren, Germany; s.hijazi@mathias-stiftung.de

**Keywords:** transperineal ultrasound, urinary continence, membranous urethra length, radical prostatectomy

## Abstract

Introduction: To predict early continence recovery following radical prostatectomy (RP) using baseline demographic and clinical data, as well as dynamic transperineal ultrasound (TPUS) parameters of membranous urethral length (MUL). Patients and Methods: A retrospective CHECK-MUL (check of membranous urethral length) study was conducted. We evaluated 154 patients who underwent RP between August 2018 and April 2023. All patients underwent pre- and postoperative dynamic TPUS to measure MUL. Urinary continence was defined as the use of one safety pad or less 3 months post surgery. The International Consultation on Incontinence Questionnaire-Short Form (ICIQ-SF) was used to assess urinary incontinence (UI). We used logistic regression to assess the association between MUL and early continence recovery. A multivariable logistic regression model was then constructed for the prediction of early continence recovery based on the MUL. Results: The median MUL observed pre- and postoperatively in this study were similar (14.6 mm and 12.9 mm). In the univariable logistic regression analysis, the pre- and postoperative MUL measured by TPUS (odds ratio (OR): 1.12; 95%-CI: 1.02–1.79; *p* = 0.05 and OR: 1.01; 95%-CI: 1.02–1.12; *p* < 0.01) directions were independent predictors of early continence recovery 3 months post surgery. In addition, age (OR: 1.23; 95%-CI: 1.11–1.42; *p* = 0.03), BMI (OR: 1.44; 95%-CI: 1.18–2.92; *p* = 0.05), and bilateral nerve sparing (OR: 1.24; 95%-CI: 1.02–1.9; *p* = 0.05) were independent predictors of urinary continence in univariable logistic regression models. Preoperative MUL >15 mm (95% CI 1.28–1.33; *p* = 0.03) and postoperative MUL >14 mm (95% CI 1.2–1.16; *p* = 0.05) were significantly associated with early continence recovery at 3 months post surgery. Conclusions: The likelihood of continence recovery increases with membranous urethral length and decreases with age, BMI, and lack of nerve sparing. Preoperative MUL >15 mm and postoperative MUL >14 mm were significantly associated with early continence recovery at 3 months post surgery.

## 1. Introduction

Postoperative functional problems such as urinary incontinence (UI) are a significant concern for patients undergoing radical prostatectomy (RP) for treatment of prostate cancer. Stress urinary incontinence (SUI) has also been shown to have a negative impact on the social life of affected patients [1,2]. In addition, there are also economic consequences of SUI for these individuals and for healthcare systems [3]. Early return to continence is one of the most important functional outcomes after RP. Several studies have reported that advancing age, BMI, comorbidities, previous radiotherapy for prostate cancer, and presence of preoperative lower urinary tract symptoms (LUTS) may be preoperative risk factors for SUI after RP [4,5,6,7]. Surgical techniques and surgeon experience are also associated with recovery of urinary continence after RP [8]. Understanding the anatomical structure and function of the pelvic floor helps to guide treatment selection, preoperative risk stratification prior to surgery, and for a satisfactory functional outcome of the procedure [9]. Despite an increasing understanding of the etiology of post-prostatectomy incontinence and improvements in surgical technique, incontinence rates are still substantial and vary widely between studies. Therefore, preservation of the MUL by accurate dissection of the prostatic apex during surgery is recommended for continence recovery after RP [5]. The membranous urethral length (MUL) as measured by magnetic resonance imaging (MRI) with almost 400 patients has been consistently shown to be a strong predictor of urinary continence recovery after RP [10,11,12]. A multiparametric magnetic resonance imaging (MRI) scan is increasingly being utilised for its significant role in diagnosis and staging of the disease; however, MR images to scan the MUL are expensive and time-consuming. During the last decade, dynamic TPUS imaging for pre- and postoperative anatomical assessment has increased compared to other imaging modalities. The ability to visualise movement and structures of the male lower urinary tract when assessing male pelvic floor muscle function has been demonstrated in men using transperineal ultrasound [13,14,15,16]. The clinical application of TPUS to reliably measure MUL and determine the degree of agreement with MRI measurements of MUL in different supine positions was investigated by Mungovan et al. [16]. The authors were able to show an excellent agreement between the TPUS and the MRI measurements. However, to our knowledge, the measurement of pre- and postoperative MUL by dynamic TPUS in patients undergoing RP to predict continence recovery has not been examined. We evaluated predictors of continence recovery following radical prostatectomy, using only those clinical and dynamic TPUS variables known preoperatively.

## 2. Materials and Methods

In the present study, we analysed 154 consecutive patients with clinically localised prostate cancer who underwent peri- and postoperative measurement of the MUL by dynamic TPUS and RP between August 2018 and April 2023. During the 4-year period, there were 154 patients in two analysed urological departments from whom we were able to collect the International Consultation on Incontinence Questionnaire-Short Form (ICIQ-SF) [17] at least preoperatively and 3 months after RP. All patients undergoing radical prostatectomy during this period with documented preoperative continent micturition and at least 3 months of follow-up were included in this study. All patients had no documented stress urinary incontinence before surgery, and only two patients had urge incontinence due to lower urinary tract symptoms with obstructive prostate. RP was performed by the conventional retropubic open approach or robotic-assisted laparoscopic prostatectomy. Pelvic lymph node dissection also was performed in all patients according to the D’Amico criteria.

According to a consensus among the investigators, the following risk factors were selected for further analysis: increased age, BMI, prostate volume, positive surgical margin sparing, and MUL. All patients were informed of the planned investigations, and TPUS was performed prior to transrectal measurement of prostate volume. Patients were excluded from the analysis if they had received neoadjuvant or adjuvant radiotherapy, incontinence, urinary retention, or previous operations on the urethra prior to RP. The criteria for bilateral nerve-sparing surgery were cT1–2a and Gleason score ≤ 3 + 4 = 7. Detailed medical history, patient characteristics including BMI, age, preoperative prostate specific antigen (PSA), prostate volume measured by transrectal ultrasound (TRUS), digital rectal exam (DRE), Gleason grading group, tumor characteristics—clinical tumor stage (cT), ISUP grade group, and dynamic TPUS were obtained for all patients.

### 2.1. Outcome Assessment and Follow Up

Urinary tract function was assessed using the International Prostate Symptom Score (IPSS) [18] at the postoperative visit after catheter removal and 3 months postoperatively. Pre- and postoperative SUI was assessed using voiding diaries, the validated ICIQ-SF questionnaires, and the patient-reported daily pads completed and recorded by the physician and patient or proxy at each visit. The primary outcome of our MUL study was to evaluate predictors of early continence recovery after radical prostatectomy between the groups. Secondary outcomes of the study were to evaluate the association between MUL and early continence recovery between the groups. Patients were defined as continent if they used “no pad” or “one safety pad” per day at 3 months postoperatively. All included patients completed a pre- and postoperative questionnaire including the ICIQ-SF questionnaire. Our primary and secondary outcomes were assessed 3 months after radical prostatectomy.

### 2.2. Transperineal Ultrasound (TPUS)

The TPUS scan was performed and interpreted at each centre by an experienced urologist (B.B, S.H) with at least 3 years of board certification. The examination was performed in the side-lying position with the spine in a neutral position and the knees flexed to 70–90 degrees and hips slightly rotated outwards. Evaluation of the pelvis was performed with TPUS as a 2D diagnostic method on a commercially available ultrasound machine (BK Ultrasound Systems and Toshiba) using a low-frequency transabdominal transducer 3.5–5 MHz. The abdominal transducer was placed on the perineal area, between the scrotum and anus in a sagittal orientation, to obtain images of the pubic symphysis, bladder, bladder neck, and urethra. Dynamic TPUS was performed in the left lateral decubitus position with the legs pulled up to the chest and a bladder filling of approximately 300 mL. The symphysis pubis is used as a stable landmark to assess bladder neck position and mobility. During the examination we were able to assess the MUL, urethral angle, and voluntary contraction of the pelvic floor.

### 2.3. Measurement

Pre- and postoperative MUL was measured using a 2D transabdominal transducer in the mid-sagittal plane as the distance from the prostate apex to the level of the urethra at the penile bulb. To visualise the pelvic floor structures (prostate apex, membranous urethra, and penile bulb) during dynamic TPUS, patients were asked to perform maximal contractions of the pelvic floor muscles and then relax.

For final analysis, the mean length of three measurements was used. The position of the bladder neck was determined at rest, during the Valsalva manoeuvre and during pelvic floor contraction. The urethral angle was measured from the anterior wall of the penile urethra to the anterior wall of the bulbar urethra at rest, during Valsalva manoeuvre and during maximal contraction.

The final length of the membranous urethra was determined by TPUS before deletion and after catheter removal on days 6–10. The percentage change in MUL was calculated as [(pre MUL-post MUL) × 100]/pre MUL and reported in the study (Figure 1).

## 3. Statistical Data Evaluation

We constructed a predictive model using MUL measurement and clinical parameters to predict early continence recovery. Continuous variables are expressed as means (standard deviation, range) and compared by independent *t*-test. Categorical variables are presented as absolute values (%) and compared using the chi-squared test or Fisher’s exact test. Univariate analysis was performed using *t*-test, analysis of variance, chi-squared, and Fisher’s exact test between the continence and incontinence groups with respect to preoperative clinical factors including patient age, BMI, estimated prostate volume, clinical stage, and neurovascular bundle sparing. Multivariate analysis was performed using a logistic regression model, and significance was tested using a likelihood ratio test. Univariable and multivariable logistic regression analyses were used to assess predictive factors with stress urinary incontinence 3 months post surgery.

To further evaluate the effect of preoperative, postoperative, and percentage change in MUL on stress urinary incontinence post surgery, multivariate logistic regression analyses were performed. Statistical analysis of the data was performed using SPSS (version 22.0, IBM Armonk, New York, USA).

## 4. Results

Clinical data for 154 patients were evaluated who met the inclusion criteria and for whom follow-up data available postoperatively were included in this study and are summarised in Table 1. In the overall cohort, median age was 65.7 years (standard deviation (SD): 6.1), preoperative prostate-specific antigen (PSA) was 7.0 ng/mL (SD 1.3), and median prostate volume was 29.8 mL (SD 7.5) (Table 1). The median MUL observed pre- and postoperatively in this study were similar (14.6 mm and 12.9 mm). Unfavorable characteristics, such as ISUP 4/5, were present in 18 (27%) patients in the overall cohort, respectively. The median length of postoperative catheterisation and hospital stay was 7 days (SD 5–9d). Bilateral nerve sparing was performed in 101 patients (66.9%). A total of 21 patients (13.6%) had unilateral nerve sparing and 30 patients (19.5%) had no nerve sparing. Postoperative extracapsular extension was found in 14 patients (9%) and positive surgical margins in 32 patients (20.8%). Overall, 14 patients (9%) had aggressive pathological features (namely, ≥pT3b and/or LNI and/or grade group 4–5 at final pathology). Histopathological data are shown in Table 1. A statistically significant difference was found, when comparing the distribution of pathologic features in patients who had a positive surgical margin (PSMs) with those who did not. In particular, PSMs were more frequent in patients with pathological stage ≥pT3b and lymph node invasion at final pathology (*p* < 0.001).

All patients were divided into two groups (continence versus incontinence according to early urinary continence 3 months after surgery); mean age at surgery (65.3 vs. 73.8 years), BMI (26.8 vs. 31.1 kg/m^2^), bilateral neurovascular bundle sparing (77 vs. 26%), preoperative MUL (15.7 vs. 13.8 mm), and postoperative MUL (13.0 vs. 10.1 mm) differed significantly (Table 2).

### 4.1. Univariable and Multivariable Logistic Regression Models

In the univariable logistic regression analysis, the pre- and postoperative MUL measured by TPUS (odds ratio (OR): 1.12; 95%-CI: 1.02–1.79; *p* = 0.05 and OR: 1.01; 95%-CI: 1.02–1.12; *p* < 0.01) directions were independent predictors of early continence recovery 3 months post surgery. In addition, age (OR: 1.23; 95%-CI: 1.11–1.42; *p* = 0.03), BMI (OR: 1.44; 95%-CI: 1.18–2.92; *p* = 0.05), and bilateral nerve sparing (OR: 1.24; 95%-CI: 1.02–1.9; *p* = 0.05) were independent predictors of urinary continence in univariable logistic regression models (Table 3). Similarly, in the separate multivariable test for independent predictors of early continence recovery, length of MUL, young age, and BMI were associated with continence recovery 3 months post surgery (Table 3).

### 4.2. Association between MUL and Early Continence Recovery

To evaluate the diagnostic performance and impact of pre- and postoperative MUL measured by TPUS in correctly predicting continence recovery, multivariate logistic regression analysis was performed adjusted for age.

Preoperative MUL >15 mm (95% CI 1.28–1.33; *p* = 0.03) and postoperative MUL >14 mm (95% CI 1.2–1.16; *p* = 0.05) were significantly associated with early continence recovery at 3 months post surgery (Table 4).

In our analysis, adjusting for the type of approach (RARP, ORP), BMI, age, and bilateral nerve sparing produced no significant effect at 3 months on our findings (Figure 1).

## 5. Discussion

Following the 15-year analysis of the ProtecT study, concerns were expressed that the assigned RP did not show a survival benefit in comparison with active surveillance or radiotherapy [19]. Stress urinary incontinence is one of the most feared complications after RP, as it is an adverse outcome that can severely affect quality of life and affect many patients. Anatomical support, pelvic innervation of the pelvis, and surgical techniques appear to be important factors in the early recovery of continence after RP [20,21]. Identifying possible predictors of early continence recovery could therefore provide physicians with prognostic information to aid clinical judgement and decision making, as well as reassuring patients. A preoperative model to predict early urinary continence based on known preoperative risk factors has been elusive. A previous study suggested that pelvic anatomy has led to a better understanding of the mechanism of urinary continence [5]. Other previous studies presented that increasing age, increasing BMI, preoperative comorbidities, adjuvant or neoadjuvant radiotherapy for prostate cancer, presence of preoperative LUTS, and decreased membranous urethral length may be preoperative risk factors for UI after RP [6,22,23,24]. Measurement of MUL in these reported studies was performed using MRI to predict continence recovery in patients after RP [6,22,23,24]. The role of RP with longer MUL using TPUS is not yet clear. TPUS measurement of MUL is reliable and not as resource-intensive as MRI, which has excellent agreement with MRI for measuring MUL in patients before RP [16]. In the present study, we used pre- and postoperative dynamic TPUS to predict continence recovery in patients after RP. Inclusion of the dynamic TPUS parameters of membranous urethral length further improves the discrimination of our model for early continence. We also showed that MUL on preoperative dynamic TPUS is an independent predictor of continence recovery after RP, after adjustment for age, BMI, and bilateral neurovascular bundle sparing. Our results showed that patients with longer pre- and postoperative MUL (15.7, 14.0 mm) as assessed by TPUS were significantly associated with early recovery of urinary continence, indicating that residual MUL also influences early continence recovery after RP (*p* < 0.001). Our results are similar to previous studies using MRI, which showed that urinary continence was achieved (68%) after RARP in patients with longer MUL, respectively [7]. Another study showed 1-month urinary continence rates of approximately 35% to 65% and 3-month rates of approximately 65% to 85% in patients with longer MUL [25]. Despite these excellent results, our study has several limitations. First, our study did not allow for the interpretation of pathways of association. Second, a major limitation of the present study was the retrospective design and the relatively small sample size. Regarding MUL measuring, we evaluated MUL by several urologists at the preoperative conference. Therefore, data on MUL summarised by a urologist blinded to clinical data would be more reproducible and less prone to selective bias in this study. As the technical skills in measuring dynamic MUL are operator-dependent, it is difficult to validate the predictive effect of technical changes between different investigators. In the future, artificial intelligence will greatly assist in such measurements and reduce the discrepancies between investigators.

Finally, a longer 12-month follow-up after RP would have been more appropriate to assess outcomes, but it was important for us to investigate predictors of early continence recovery. To our knowledge, the measurement of pre- and postoperative MUL by TPUS in patients undergoing RP to predict early continence recovery has not been previously investigated. The TPUS can be reliably measured by urologists and may further facilitate a patient-tailored approach to radical treatment of prostate cancer.

Further high-quality multicentre studies evaluating the measurement of pre- and postoperative MUL by TPUS with long-term follow-up are needed to provide further evidence.

## 6. Conclusions

In conclusion, age, BMI, bilateral nerve sparing, and membranous urethral length would be predictive factors for early continence recovery at 3 months following RP. Preoperative MUL >15 mm and postoperative MUL >14 mm were significantly associated with early continence recovery at 3 months post surgery.

A further benefit of this model is the identification of a cohort of patients at high risk of UI who may be suitable for perioperative pelvic floor muscle training to improve urinary continence.

## Figures and Tables

**Figure 1 diagnostics-14-00853-f001:**
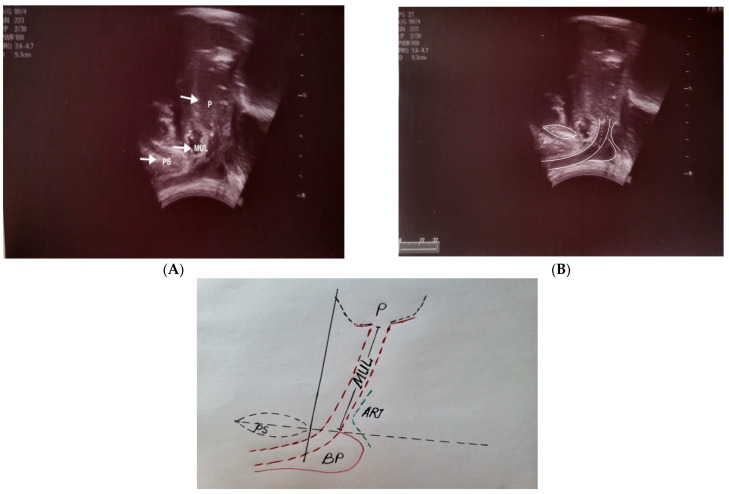
Transperineal ultrasound (TPUS), showing mid-sagittal planes (**A**,**B**) using a low-frequency transabdominal transducer 3.5–5 MHz. P: prostate, MUL: membranous urethral length, PS: pubic symphysis, BP: bulb of the penis, ARI: ano-rectal junction.

**Table 1 diagnostics-14-00853-t001:** Baseline demographic and clinical characteristics of all included patients, stratified according to surgical approach.

Variable	[All Cases] *n* = 154	Open RP *n* = 71	RARP *n* = 83	*p*-Value
Age (years), median (SD)	65.7 (6.1)	66.4 (6.1)	65.2 (6.1)	0.6
BMI (kg/m^2^), median (SD)	27.7 (3.1)	27.7 (2.9)	27.6 (3.2)	0.68
PSA (ng/mL), median (SD)	7.0 (1.3)	7.0 (1.3)	7.1 (1.3)	0.56
Prostate volume (mL), median (SD)	29.8 (7.5)	30.9 (8.2)	27.9 (6.7)	0.09
ASA score, *n* (%)				0.53
1–2	103 (66.8)	44 (61.9)	59 (71)	
3	51 (33.1)	27 (38)	24 (28.9)	
Pathological ISUP Score, *n* (%)				0.67
ISUP1	40 (25.9)	20 (28.1)	20 (24.1)	
ISUP2	82 (53.2)	34 (47.8)	48 (57.8)	
ISUP3	14 (9.1)	7 (9.8)	6 (7.2)	
ISUP4	10 (6.4)	6 (8.4)	5 (6.0)	
ISUP5	8 (5.2)	4 (5.6)	4 (4.8)	
pT-stage, *n* (%)				0.14
≤pT2	141 (91.5)	64 (90.1)	77 (92.7)	
pT3	11 (7.1)	5 (7.0)	6 (7.2)	
pT4	3 (1.9)	3 (2.8)	0 (0.0)	
pN-stage, *n* (%)				0.2
pN0	140 (13.7)	62 (11)	78 (16.2)	
pN1	14 (9.0)	6 (8.4)	8 (9.6)	
Neurovascular bundle spaing, *n* (%)				0.65
No	30 (19.5)	19 (26.8)	11 (13.3)	
yes, bilateral	103 (66.9)	47 (66.2)	56 (67.5)	
yes, unilateral	21 (13.6)	5 (7.0)	16 (19.3)	
Gleason score, *n* (%)				
≤6	21 (13.6)	8 (11.2)	13 (15.6)	
7	98 (63.6)	40 (56.3)	58 (69.8)	
≥8	35 (22.7)	23 (32.3)	12 (14.4)	
Positive surgery margin, *n* (%)				0.92
No	122 (79.2)	56 (78.9)	66 (79.5)	
yes	32 (20.8)	17 (20.5)	15 (21.1)	
EBL, (mL)	277 ± 187 (40–1560)	480 ± 230 (210–1560)	250 ± 123 (40–710)	0.01
Operative time, min	254 ± 77 (156–380)	190 ± 86 (156–220)	235 ± 74 (197–380)	0.05
Preoperative MUL (cm), median (SD)	14.6 ± 1.4 (7.6–21.1)	14.7 ± 2.3 (13.2–20.2)	13.8 ± 1.9 (12.1–14.7)	0.71
Postoperative MUL (cm), median (SD)	12.9 ± 1.5 (6.8–17.5)	13.0 ± 2.2 (7.8–11.1)	13.6 ± 2.0 (6.9–13.8)	0.78

SD: standard deviation; BMI: body mass index; PSA: prostatic specific antigen; MUL: membranous urethra length; ISUP: international society of urological pathology; pT: pathological T category; pN: lymph nodes category; ASA: American Society of Anesthesiologists; EBL: estimated blood loss; a *p*-value is a measure of the probability that an observed difference could have occurred just by random chance (*p*-value was calculated with chi-squared test). A *p*-value less than 0.05 is statistically significant, all values are median (IQR) or frequencies (%).

**Table 2 diagnostics-14-00853-t002:** Baseline demographic and clinical characteristics of all included patients, stratified according to early continence recovery.

Variable	[All Cases] *n* = 154 (100%)	Continent *n* = 105 (68%)	Incontinent *n* = 49 (32%)	*p*-Value
Age (years), median (SD)	65.7 (6.1)	65.3 (7.2)	73.8 (4.2)	0.05
BMI (kg/m^2^), median (SD)	27.7 (3.1)	26.8 (3.9)	31.1 (4.2)	0.04
PSA (ng/mL), median (SD)	7.0 (1.3)	7.6 (3.3)	8.8 (2.4)	0.54
Prostate volume (mL), median (SD)	29.8 (7.5)	33.4 (7.9)	27.2 (6.5)	0.94
ASA score, *n* (%)				0.07
1–2	103 (66.8)	78 (74.2)	25 (51)	
3	51 (33.1)	27 (25.7)	24 (48.9)	
Pathological ISUP Score, *n* (%)				0.06
ISUP1	40 (25.9)	23 (21.9)	17 (28.5)	
ISUP2	82 (53.2)	53 (50.4)	29 (57.1)	
ISUP3	14 (9.1)	10 (13.3)	4 (8.1)	
ISUP4	10 (6.4)	8 (7.6)	2 (4.1)	
ISUP5	8 (5.2)	7 (6.6)	1 (2.0)	
pT-stage, *n* (%)				0.12
≤pT2	141 (91.5)	92 (87.6)	47 (95.9)	
pT3	11 (7.1)	9 (8.5)	2 (4.1)	
pT4	3 (1.9)	3 (2.8)	0 (0.0)	
pN-stage, *n* (%)				0.07
pN0	140 (13.7)	99 (94.2)	41 (83.6)	
pN1	14 (9.0)	6 (5.7)	8 (16.3)	
Neurovascular bundle sparing, *n* (%)				0.01
No	30 (19.5)	17 (16.1)	13 (26.5)	
yes, bilateral	103 (66.9)	77 (73.3)	26 (53.1)	
yes, unilateral	21 (13.6)	11 (10.4)	10 (20.4)	
Gleason-score, *n* (%)				
≤6	21 (13.6)	13 (12.3)	8 (16.3)	
7	98 (63.6)	76 (72.3)	22 (44.8)	
≥8	35 (22.7)	16 (15.2)	19 (38.7)	
Positive surgery margin, *n* (%)				0.06
No	122 (79.2)	76 (72.3)	46 (93.8)	
yes	32 (20.8)	29 (27.6)	3 (6.1)	
Preoperative MUL (cm), median (SD)	14.8 ± 1.7 (8.9–20.4)	15.7 ± 2.0 (12.1–20.4)	12.3 ± 1.1 (8.1–14.8)	0.01
Postoperative MUL (cm), median (SD)	13.5 ± 1.4 (5.8–18.9)	13.0 ± 1.6 (10.8–20.1)	10.1 ± 2.3 (5.9–13.8)	0.00

MUL: membranous urethra length; BMI: body mass index; PSA: prostatic specific antigen; ISUP: international society of urological pathology; pT: pathological T category; pN: lymph nodes category; ASA: American Society of Anesthesiologists; a *p*-value is a measure of the probability that an observed difference could have occurred just by random chance (*p*-value was calculated with chi-squared test). A *p*-value less than 0.05 is statistically significant.

**Table 3 diagnostics-14-00853-t003:** Univariable and multivariable logistic regression analyses for identifying predictive factors for early urinary continence.

Variables	Univariable OR (95%CI)	*p*-Value	Multivariable OR (95%CI)	*p*-Value
Age * (years)	1.23 (1.11–1.42)	0.03	1.04 (1.15–1.18)	0.02
BMI (kg/m^2^)	1.46 (1.18–2.92)	0.05	1.32 (1.19–2.77)	0.04
Prostate volume (mL)	2.61 (0.76–2.22)	0.76	2.94 (0.95–2.74)	0.82
Preoperative PSA (ng/mL) *	1.53 (0.98–1.96)	0.66	1.52 (0.97–1.96)	0.65
ASA score (1–2 vs. 3)	2.13 (0.88–4.74)	0.53	2.34 (0.97–4.89)	0.78
Pathological ISUP Score				
(≤ISUP2 vs. ≥ISUP3)	1.83 (0.38–5.1)	0.92	1.73 (0.57–4.42)	0.88
pT-stage				
(≤T2 vs. ≥T3)	1.0 (0.79–4.13)	0.11	1.0 (0.77–4.6)	0.16
Gleason-score				
≤6	1.0 (ref)		1.0 (ref)	
7	1.77 (0.55–6.3)	0.52	1.53 (0.63–4.8)	0.68
≥8	1.56 (0.82–5.8)	0.79	0.84 (0.34–3.7)	0.89
Surgical approach				
Open vs. RARP	0.83 (0.21–1.04)	0.96	0.94 (0.49–1.12)	0.99
Operation time *	1.04 (0.72–1.82)	0.63	1.01 (0.83–1.77)	0.69
EBL *	1.06 (0.95–1.55)	0.73	1.0 (0.90–1.49)	0.98
neurovascular bundle spring				
(Bilateral or unilateral versus none)	1.24 (1.02–1.9)	0.05	1.01 (1.37–1.79)	0.04
Operation time *	2.42 (0.45–3.9)	0.92	2.12 (0.55–3.7)	0.81
Preoperative MUL	1.12 (1.01–1.79)	0.05	1.13 (1.05–1.82)	0.05
Postoperative MUL	1.01 (1.02–1.12)	<0.01	1.01 (1.01–1.15)	0.00

* Continuous variables; OR: odds ratio; MUL: membranous urethra length; BMI: body mass index; PSA: prostatic specific antigen; ISUP: international society of urological pathology; pT: pathological T category; ASA: American Society of Anesthesiologists; EBL: estimated blood loss.

**Table 4 diagnostics-14-00853-t004:** Association of preoperative and postoperative TPUS-measured MUL with early continence recovery.

Variables	Unadjusted OR (95%CI)	*p*-Value	Age-Adjusted OR (95%CI)	*p*-Value
Preoperative MUL, mm ≤11 vs. >16	1.0 (1.01–1.17)	<0.01	1.04 (1.04–1.15)	0.00
Preoperative MUL, mm ≤12 vs. >15	1.28 (1.12–1.33)	0.03	1.30 (1.18–1.32)	0.05
Preoperative MUL, mm ≤13 vs. >14	1.75 (0.78–2.12)	0.61	2.0 (0.89–2.44)	0.58
Postoperative MUL, mm1.0 (1.06–1.14) ≤8 vs. >15	0.01	1.0 (1.05–1.13)		0.01
Postoperative MUL, mm ≤9 vs. >14	1.18 (1.2–1.16)	0.05	1.14 (1.08–1.21)	0.04
Postoperative MUL, mm ≤10 vs. >13	1.49 (0.99–2.1)	0.08	1.40 (0.97–2.02)	0.07
Postoperative MUL, mm ≤11 vs. >12	2.3 (0.81–2.23)	0.72	2.15 (0.92–3.4)	0.46

TPUS: transperineal ultrasound; OR: odds ratio; MUL: membranous urethra length. *p*-value is a measure of the probability that an observed difference could have occurred just by random chance (*p*-value was calculated with chi-squared test). A *p*-value less than 0.05 is statistically significant.

## Data Availability

Due to data protection, all analyses are anonymised by the corresponding author.

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
