# Peer review of "Predictors of Early Continence Recovery Following Radical Prostatectomy, Including Transperineal Ultrasound to Evaluate the Membranous Urethra Length (CHECK-MUL Study)"

_diagnostics, 2024, doi:10.3390/diagnostics14080853_

Round 1

Reviewer 1 Report

Comments and Suggestions for Authors

Welcome

Congratulations on the work.

Another risk factor/predictive. Well done.

Of course US examination has some limitations as it is a dynamic study.

We all know that on the same patients the results can be different when made by different physicians. How to solve this issue?

Comments on the Quality of English Language

Minor editing of English language required

Author Response

------------------Template Response Letter to Reviewer Comments ----------------------------

_________________________________________________________________________

Thank you for the comments on our manuscript entitled „Predictors of early continence recovery following radical prostatectomy, including transperineal ultrasound to evaluate the membranous urethra length (CHECK- MUL Study)

(Paper No. diagnostics-2909553). We appreciate the suggested modifications and have revised the manuscript accordingly. The changed and added texts in the revised manuscript are shown in yellow. The detailed responses to the reviewers’ comments are as follows:

Reviewer #1:

Comment 1: Of course, US examination has some limitations as it is a dynamic study.

We all know that on the same patients the results can be different when made by different physicians. How to solve this issue?

Response 1:

Thank you for a very important comment. It is absolutely an important point to discuss the discrepancy in measurement between the investigators. We have included this point as one of the weaknesses of this study. Artificial intelligence will be the solution in the future

As the technical skills in measuring dynamic MUL are operator-dependent, it is difficult to validate the predictive effect of technical changes between different investigators. In the future, artificial intelligence will greatly assist in such measurements and reduce the discrepancies between investigators.

Reviewer #2:

Comment 1:

1: Inclusion and exclusion criteria are not clarified.

Response 1:

Thank you very much for your interpretation. The inclusion criteria were emphasised in the text. The exclusion criteria were already listed and have been supplemented with regard to previous operations on the urethra.

Comment 2: History of diseases and surgeries especially urethral surgery in the past in the history of patients that are not specified.

Response 2:

The addition regarding the preoperatins on the urethra was listed.

None of the patients underwent surgery for urinary incontinence previously

Comment 3:

The type of urinary incontinence is unknown and has not been discussed.

Response 3: Thank you for the important information, we added the type of urinary incontinence.

Reviewer 2 Report

Comments and Suggestions for Authors

Dear Author 

This is a good and well designed manuscript .

1: Inclusion and exclusion criteria are not clarified . 

2: History of diseases and surgeries especially urethral surgery in the past in the history of patients that are not specified. 

3: The type of urinary incontinence is unknown and has not been discussed.

Author Response

------------------Template Response Letter to Reviewer Comments ----------------------------

_________________________________________________________________________

Thank you for the comments on our manuscript entitled „Predictors of early continence recovery following radical prostatectomy, including transperineal ultrasound to evaluate the membranous urethra length (CHECK- MUL Study)

(Paper No. diagnostics-2909553). We appreciate the suggested modifications and have revised the manuscript accordingly. The changed and added texts in the revised manuscript are shown in yellow. The detailed responses to the reviewers’ comments are as follows:

Reviewer #2:

Comment 1:

1: Inclusion and exclusion criteria are not clarified.

Response 1:

Thank you very much for your interpretation. The inclusion criteria were emphasised in the text. The exclusion criteria were already listed and have been supplemented with regard to previous operations on the urethra.

Comment 2: History of diseases and surgeries especially urethral surgery in the past in the history of patients that are not specified.

Response 2:

The addition regarding the preoperatins on the urethra was listed.

None of the patients underwent surgery for urinary incontinence previously

Comment 3:

The type of urinary incontinence is unknown and has not been discussed.

Response 3: Thank you for the important information, we added the type of urinary incontinence.